# Plant-Based, Antioxidant-Rich Snacks Elevate Plasma Antioxidant Ability and Alter Gut Bacterial Composition in Older Adults

**DOI:** 10.3390/nu13113872

**Published:** 2021-10-29

**Authors:** Jing-Yao Zhang, Hui-Chen Lo, Feili Lo Yang, Yi-Fang Liu, Wen-Mein Wu, Chi-Chun Chou

**Affiliations:** 1Department of Nutritional Science, Fu Jen Catholic University, Zhongzheng Rd., Xinzhuang Dist., New Taipei City 24205, Taiwan; zhangjingyao2012@126.com (J.-Y.Z.); 031765@mail.fju.eud.tw (F.L.Y.); 070647@mail.fju.edu.tw (Y.-F.L.); 050582@mail.fju.edu.tw (W.-M.W.); 2Ph.D. Program in Nutrition and Food Science, Fu Jen Catholic University, Zhongzheng Rd., Xinzhuang Dist., New Taipei City 24205, Taiwan; 3Department of Otolaryngology, Yonghe Cardinal Tien Hospital, Zhongxing St., Yonghe Dist., New Taipei City 23445, Taiwan; cthyh10680@gmail.com

**Keywords:** antioxidant enzymes, gut microbiota, older adults, plant foods, total antioxidant capacity

## Abstract

Plant-rich diets alleviate oxidative stress and gut dysbiosis and are negatively linked to age-associated chronic disorders. This study examined the effects of consuming plant-based, antioxidant-rich smoothies and sesame seed snacks (PBASS) on antioxidant ability and gut microbial composition in older adults. Healthy and sub-healthy older adults (*n* = 42, 79.7 ± 8.6 years old) in two senior living facilities were given PBASS for 4 months. Blood and fecal samples were collected from these individuals at the baseline and after 2 and 4 months of PBASS consumption. After 2 months, serum levels of albumin and high-density lipoprotein-cholesterol and the ratio of reduced to oxidized glutathione (GSH/GSSG) had increased significantly and erythrocytic glutathione, GSH/GSSG and superoxide dismutase activity had decreased significantly compared with baseline levels (*p* < 0.05). After 4 months, red blood cells, hematocrit, serum blood urea nitrogen and erythrocyte glutathione peroxidase activity had decreased significantly, whereas plasma and erythrocyte protein-bound sulfhydryl groups had increased significantly. Furthermore, plasma glutathione and total antioxidant capacity were significantly greater after 2 months and increased further after 4 months of PBASS consumption. The results of next generation sequencing showed that PBASS consumption prompted significant decreases in observed bacterial species, their richness, and the abundance of Actinobacteria and Patescibacteria and increases in Bacteroidetes in feces. Our results suggest that texture-modified, plant-based snacks are useful nutrition support to benefit healthy ageing via the elevation of antioxidant ability and alteration of gut microbiota.

## 1. Introduction

Increased oxidative and inflammatory stress levels contribute to the development of age-associated chronic disorders (AACD), such as cancer; diabetes; muscular atrophy; and cardiovascular, pulmonary and neurological diseases [1]. In Taiwan, population aging has been accentuated by zero population growth, and Taiwan will consequently become a super-aged society in which over 20% of the population is in the age group of 65+ years by 2026 [2]. The rapid growth of an aging population leads to increased demands for medical and long-term care, necessitating a focus on “healthy ageing” to delay the development and progression of AACD [3]. Studies have found that diets that include a variety of vegetables, fruits, legumes, and nuts are rich in antioxidant vitamins, phytochemicals, monounsaturated fatty acids, and dietary fiber, and have anti-inflammatory effects against aging-related disorders [4].

The findings of a cross-sectional survey of healthy women showed that greater dietary diversity, with a focus on vegetables, fruits, and milk and dairy products is associated with reduced oxidative stress, indicated by increased total antioxidant capacity (TAC), superoxide dismutase (SOD), and glutathione peroxidase (GPx) [5]. Using telomere length as an indicator of aging, one study found that Chinese women who consumed greater amounts of plant foods had longer telomere lengths and lower serum levels of the inflammatory marker C-reactive protein (CRP) [6]. In addition, high fecal levels of fiber-degrading bacteria, such as Firmicutes and Bacteroidetes, and short-chain fatty acids (SCFA), which are important metabolites for maintaining intestinal health, have been found in healthy subjects whose diets are largely plant-based [7]. However, chewing and swallowing dysfunction, sensory changes, and medication use among older adults may result in changing dietary preferences and poor appetite [8], which lead to markedly reduced consumption of plant-based foods and an increased risk of undernutrition [9]. Moreover, studies have found an association between AACD and an altered dietary pattern characterized by low levels of vegetable and fruit consumption and dietary diversity, which leads to elevated oxidative stress and imbalanced gut microbial composition [5,10].

Strategies deployed to increase the consumption of plant-based foods in older adults with chewing and/or swallowing problems include the modification of food texture and the enrichment and reshaping of food items [11]. To the best of our knowledge, no study has investigated the effects of plant-based snacks on oxidative status and gut microbiota in healthy and sub-healthy older adults, who comprise over 70% of the older population [12]. In this study, we hypothesized that the consumption of diverse, plant-based, texture-modified snacks could reduce oxidative stress and alter gut bacterial composition in older adults. Therefore, we investigated the effects of consuming plant-based, antioxidant-rich smoothies and snacks (PBASS) on blood antioxidant ability and gut microbial composition in healthy and sub-healthy older adults. We also explored the relationship between antioxidant ability and gut microbes in this group.

## 2. Materials and Methods

### 2.1. Study Participants and Ethics

Healthy and sub-healthy older adults (*n* = 59, 15 men and 44 women) were recruited in two senior living facilities operated by Yonghe Cardinal Tien Hospital in New Taipei City, Taiwan. Inclusion criteria for the study were an age of 65 years and above, residence in the facilities for over 2 months, and a healthy or sub-healthy diagnosis by physicians. The sub-healthy group was defined according to the following characteristics: no existing pathological condition; existing prehypertension, overweight or underweight, serum blood lipids above the borderline for high lipid content, or suboptimal renal or hepatic health, as diagnosed through a routine health exam [13]. Exclusion criteria were diagnosed cancer, chronic obstructive pulmonary disease, severe disability, or dementia. Residents of these two facilities were invited to attend a briefing aimed at recruiting participants for the study, and personal visits were arranged for potential participants with the project specialists to provide information about the project.

Ethical approval to perform this study was obtained from the institutional review board of Fu Jen Catholic University (FJU-IRB-C106019). Signed informed consent forms were obtained from the participants before the study commenced.

### 2.2. Study Design

The participants’ personal information on their ages and disease histories were collected, and their anthropometric measurements, namely body weight, height, and waist and hip circumferences, were recorded during the first personal interview (at the baseline). Their body mass indexes (BMI) and ratios of waist to hip circumference (WHR) were subsequently calculated.

After the first personal interview, all participants were requested to consume PBASS daily along with the regular meals provided by the institution for a 4-month period. The regular meals of the 2 senior living facilities were designed by one dietician and provided to all participants as usual. PBASS was delivered weekly in a cold storage bag to each participant; the smoothies were stored in the freezer and the sesame seed snacks were left at room temperature in the participants’ rooms. The consumed quantities of PBASS were weighed weekly by project specialists. During the interviews held at the baseline and after 2 and 4 months, the participants were asked about their consumption of regular meals. Anthropometric measurements and fasting blood and fecal samples were also collected at this time.

### 2.3. Composition of Plant-Based Antioxidant Smoothies and Sesame Seed Snacks

According to the results of the Nutrition and Health Survey (2013–2016) implemented in Taiwan, decreased chewing ability results in lower intakes of vegetables, fruits, and nuts among older adults [8]. Therefore, we designed four different types of plant-based smoothies with different flavors and colors: orange, green, dark green, and purple, to increase the diversity of plant sources used in the smoothies and avoid monotony resulting from continual consumption of the same smoothies.

All of the smoothies were prepared by a contracted food manufacturer. Procedures for preparing these plant-based smoothies comprised the following processes: cleaning, pre-preparation, blanching, micro-processing, packaging, and frozen storage at −30 °C. The total bacterial numbers and total coliform bacteria and E. coli counts of the frozen smoothies were monitored for a period of 24 months and complied with the microbiological hygienic standards for frozen food of the Food and Drug Administration, Ministry of Health and Welfare, Taiwan. Each serving of a plant-based smoothie contained 1 exchange of vegetables (2 kinds), 1 exchange of fruits (2 kinds), and 1 exchange of nuts, which provided 145–186 kcals, 21–29 g of carbohydrate, 4–5 g of protein, 7 g of lipid, 2.7–4.7 g of dietary fiber, and varying amounts of minerals, vitamins, and phytochemicals [14]. These plant-based smoothies contained low levels of saturated fatty acids, zero cholesterol, and low levels of sodium and sugar, with each serving providing 10–15% of the DRIs of vitamins A, C, and B1 and potassium, calcium, and magnesium in addition to being rich in phytochemicals [14]. The ingredients, calories, and the contents of macronutrients and dietary fiber of the four plant-based smoothies are listed in the Appendix A.

To increase intakes of calcium, polyunsaturated fatty acids, vitamin E, and phytochemicals, sesame seed snacks were also provided. Each serving of sesame seed powder and spread, respectively, provided 57.6 and 63.3 kcal, 2.1 g and 2.5 g of carbohydrate, 1.6 g and 1.3 g of protein, 5.5 g and 5.7 g of lipids, and varying quantities of vitamins and minerals [14]. All participants were provided with 5 servings of plant-based smoothies (1 serving of each type plus 1 serving of any type, 150 g in each serving) and 3 servings of sesame seed snacks (powder or spread) per week (10 g in each serving). Participants received these PBASS for a 4-month period. The individual compliance rates relating to PBASS consumption were recorded and calculated.

### 2.4. Serological and Biochemical Analyses

Fasting blood samples were collected from the participants to obtain whole blood, serum, plasma, and erythrocytes. Plasma and erythrocyte samples were stored at −80 °C prior to performing further analyses. For the whole blood samples, complete blood counts, including white blood cells, red blood cells (RBCs), hemoglobin, hematocrit, and platelets were measured using a hematology analyzer (GEN-S System 2, Beckman Coulter, Miami, FL, USA). Serum concentrations of albumin, triglycerides, cholesterol, low-density lipoprotein-cholesterol (LDL-C), high-density lipoprotein-cholesterol (HDL-C), blood urea nitrogen (BUN), creatinine, and high-sensitivity C-reactive protein (CRP) were measured using an automatic clinical chemistry analyzer (Hitachi 747, Tokyo, Japan).

### 2.5. Indicators of Oxidative Status in the Plasma and Erythrocytes

To determine the oxidative status of older adults, the products of lipid peroxidation, the levels of antioxidants, and the activities of antioxidant enzymes were identified using a previously described procedure [15]. Specifically, we identified thiobarbituric acid reactive substances (TBARS), TAC, reduced glutathione (GSH), oxidized glutathione (GSSG), total sulfhydryl groups (TSH), protein-bound sulfhydryl groups (PBSH), and non-protein sulfhydryl groups (NPSH) in the plasma and erythrocytes. The status of the enzymatic antioxidant defense system within the erythrocytes, as indicated, for example, by the activities of SOD, GPx, and catalase were measured [15].

### 2.6. Short Chain Fatty Acids in the Feces

After participants had defecated, fecal samples, uncontaminated by urine or toilet water, were collected immediately into S-Y feces test bottles (Shin-Yung Medical Instruments Co., Ltd., Taipei, Taiwan). The samples were stored at 4 °C in the nursing station and transferred to a −80 °C freezer within 24 h. SCFA content in the fecal samples, notably acetic acid (AA), propionic acid (PA), and butyric acid (BA) were determined used the high-performance liquid chromatography-ultraviolet (HPLC-UV) method developed by De Baere et al. [16]. In brief, the fecal sample was diluted in sterile water, extracted by diethyl ether, and acidified using concentrated hydrochloric acid. The aqueous phase of the liquid-liquid extraction was transferred to the HPLC-UV apparatus (Hitachi High-Tech, Tokyo, Japan) and analyzed at a wavelength of 210 nm. SCFAs were separated using a C18 Hypersil Gold aQ column (4.6 mm × 150 mm i.d., 3 μm) with a guard column (4.0 mm × 10 mm, 3 μm) (Thermo Fisher Scientific Inc., Branchburg, NJ, USA) at 30 °C. Mobile phase A comprised 20 mM NaH_2_PO_4_ (pH 2.2) and mobile phase B comprised 100% acetonitrile. A gradient elution program was used following the procedure described by De Baere et al. [16] To correct for variations during the preparation and analysis procedures, succinic acid was used as the internal standard. The amounts of SCFAs were calculated with references to calibration curves using 0.5, 1, 2.5, 5, 10, 25, and 50 mM of AA, PA, and BA after correcting with the recovery rate of succinic acid.

### 2.7. Gut Microbiota Composition

Total bacterial DNA was isolated from feces by an EasyPrep Stool Genomic DNA kit (Biotools, New Taipei City, Taiwan) according to the manufacturer’s instructions. The concentration of extracted genomic DNA was determined by measuring the absorbance at 260 nm and 280 nm using the NanoDrop ND-2000 spectrophotometer (Thermo Scientific, Waltham, MA, USA). DNA concentration and purity were confirmed with 1% agarose gel electrophoresis.

Purity confirmed DNA (1 ng/μL) was used to amplify the V3–V4 hypervariable region of a bacterial 16S rRNA gene with the forward primer 391F (5′-CCTACGGGNGGCWGCAG-3′) and the reverse primer 806R (5′-GACTACHVGGGTATCTAATCC-3′) in the polymerase chain reaction (PCR). The PCR conditions were a pre-denaturation cycle performed at 95 °C for 3 min, 25 cycles of denaturation at 95 °C conducted for 30 s, annealing conducted at 55 °C for 30 s, elongation conducted at 72 °C for 30 s, and a final post-elongation cycle performed at 72 °C for 5 min. The PCR products were confirmed by applying 2% agarose gel electrophoresis to detect 400–450 base pairs of DNA. After the PCR products had been purified using a Qiagen Gel Extraction kit (Qiagen, Hilden, Germany), they were used to construct the libraries and paired-end sequenced (2 × 150 bp) using a NovaSeq 6000 system (Illumina, San Diego, CA, USA) at the Biotools Co., Ltd. (New Taipei City, Taiwan).

FLASH software (V1.2.11, Institut Pasteur, Paris, France), QIIME software (v1.9.1) [17], the UCHIME algorithm, and the Gold Database (16S) were used to filter and merge raw sequencing data, and the chimera sequences were removed to obtain effective tags. Uparse software (Uparse v7.0.1090) was used to cluster all effective tags, and tags with sequence identities greater than 97% were assigned to the same operational taxonomic units (OTUs). The representative sequence for each OTU was annotated and assigned to a specific species using the RDP Classifier (v. 2.2) and PyNAST (v. 1.2) programs, the GreenGenes database (gg_13_8; default) and the Silva database (132; 2017.12), as described in the study of Lu et al. [18].

Microbial diversity was assessed within samples (alpha-diversity) and time points (beta-diversity). The alpha-diversity value of each sample was calculated on the basis of the observed OTUs, the abundance-based coverage estimator (ACE), the bias-corrected Chao richness estimator (Chao1), and the Shannon and Simpson indexes using QIIME (v. 1.9.1). Beta-diversity measures were calculated according to the UniFrac distance, and the UPGMA algorithm was used to perform principal coordinate analysis (PCoA) to visualize similarities among the samples using the ade4 and ggplot2 R packages and QIIME (v1.9.1) [19].

### 2.8. Statistical Analyses

Statistical analyses were performed using SAS software (version 9.4, SAS Institute, Inc., Cary, NC, USA). Data are expressed as the mean ± standard deviation (SD). After adjusting for age, gender, and PBASS compliance, generalized estimating equations with repeated measure analysis were performed to determine changes in the data for the total participants during the implementation of the experiment. The following data were analyzed: anthropometric data, blood counts, serum biochemical parameters, antioxidant levels in plasma and erythrocytes, lipid peroxidation products, antioxidant enzyme activities, fecal levels of SCFAs, alpha-diversity indices, and gut microbe percentages. Logarithmic transformation was completed for the percentages of gut microbes because these values did not show a normal distribution. Where there was a significant difference (*p* < 0.05), Tukey’s HSD test was used to compare the means of these parameters at the baseline and after 2 and 4 months of PBASS consumption.

A post hoc power and effect size calculation was conducted using a statistical power analysis program G*Power 3.1 on the bases of an F test, fixed effects of ANOVA and a priori analysis at α = 0.05 [20].

To evaluate the effect of time relating to PBASS consumption, changes of these parameters from the baseline to 2 and 4 months, respectively, after PBASS consumption commenced were calculated and compared using Student’s t test. We applied a forward stepwise multiple linear regression analysis and considered the abundance of fecal bacteria with significant differences at the baseline and after 2 and 4 months of PBASS consumption (*p* < 0.05) as the dependent variables for predicting TAC levels in the plasma and erythrocytes.

We applied the Agricolae R package with Kruskal–Wallis test to compare differences in beta-diversity according to the time duration at the baseline and after 2 and 4 months of PBASS consumption. Linear discriminant analysis (LDA) combined with effect size measurements (LEfSe) was conducted to identify taxonomic biomarkers according to relative abundance, considered at the taxonomic levels of genus or higher at the 3 time points. The threshold for the logarithmic LDA score for discriminative features was set to 3.0 to identify significant differences in taxa abundance (*p* < 0.05).

## 3. Results

### 3.1. Anthropometric Measurements

Of the 59 older adults recruited for this study, 30 women and 12 men participated fully in the PBASS intervention and blood sample collection process; however, 1 woman and 1 man refused to provide fecal samples after 4 months of PBASS consumption (Figure 1). The female subjects (*n* = 30, 77.7 ± 7.6 years old) were younger than the male subjects (*n* = 12, 85.3 ± 8.4 years old); however, there were no significant differences in body mass index (BMI) and the ratio of waist to hip circumference (WHR) between female and male subjects. The compliances of consuming smoothies and sesame snacks were 88.0% and 92.5%, respectively.

After adjustments were made for age, gender, and PBASS compliance, the results of generalized estimating equations combined with repeated measures analysis showed that body weight increased significantly after 2 months (Table 1). To assess the time effects of PBASS consumption, we compared changes in each of the parameters within 2- and 4-months of the intervention and found no significant difference in body weight changes. The levels and changes of BMI, waist and hip circumference and WHR were not significantly altered by PBASS consumption.

### 3.2. Complete Blood Counts and Serum Biochemical Parameters

Levels of RBC, hemoglobin and hematocrit in males and females were within the low, borderline and normal ranges for adults. After adjustments were made for age, gender, and PBASS compliance, RBC and hematocrit levels decreased significantly, whereas hemoglobin levels increased significantly after 4 months of PBASS consumption (Table 2). To assess the time effects of PBASS consumption, we found that the decreases in RBC and hematocrit were greater after 4 months than 2 months. The levels and changes of white blood cells and platelets were not significantly altered by PBASS consumption. In serum biochemical parameters, albumin and BUN levels had decreased significantly, and HDL-C levels had increased significantly after 4 months of the intervention (Table 2). In addition, the increase in HDL-C levels and the decrease in BUN levels were greater in older adults after 2 and 4 months, respectively.

### 3.3. Oxidative Status in the Plasma

Table 3 shows the levels of the antioxidant indices and TBARS in the plasma. GSH and TAC levels in the plasma increased significantly after 2 months compared with baseline levels and continued to increase after 4 months. Plasma GSSG levels were significantly lower after 2 months relative to baseline levels. GSH/GSSG was notably higher after 2 and 4 months compared with baseline levels. In addition, increases in the plasma GSH, TSH, PBSH, and TAC after 4 months and an increase in GSH/GSSG along with a decrease in GSSG after 2 months were appreciable.

### 3.4. Oxidative Status in the Erythrocytes

Following adjustments, GSH, GSSG, GSH/GSSG and SOD activity in the erythrocytes decreased significantly and PBSH increased significantly after 2 months compared with baseline levels, and GSH levels decreased further after 4 months (Table 4). Levels of TSH increased significantly and GPx activity decreased significantly after 4 months compared with baseline levels. The decrease in erythrocyte GPx activity was greater after 4 months than 2 months. Changes in TBARS, TAC, and the activities of SOD and catalase in the erythrocytes were not significantly different at the 2- and 4-month time points of the intervention.

### 3.5. Short-Chain Fatty Acid Levels in the Feces

The fecal levels of SCFAs were not significantly altered by PBASS during the 4 months of the intervention (Table 5). Changes in SCFA content in the feces were not significantly different after 2 and 4 months of PBASS consumption.

### 3.6. Fecal Bacterial Composition

The number of bacterial species, that is, observed OUTs and species richness (Chao1 and ACE) decreased significantly after 2 and 4 months compared with the baseline values (Table 5). Changes in observed OTUs, Chao1, and ACE estimators were not markedly different at the 2- and 4-month time points. There were no notable differences in the levels of species diversity, that is, the Shannon and Simpson indices during the experimental period.

To evaluate the effects of the PBASS intervention on the composition of gut microbiota, the PCoA score plot with 3 separated clusters was used to show beta-diversity at the baseline and 2 and 4 months after the intervention was initiated. The overlapping clusters indicated that beta-diversity in older adults was not significantly altered after 4 months (Figure 2a).

At the phylum level, the top ten microorganism populations were analyzed. Firmicutes, Bacteroidetes, Proteobacteria, and Verrucomicrobia, accounting for over 95% of the total bacterial number, were four common dominant phyla at the baseline and after 2 and 4 months of the intervention (Figure 2b).

To identify the effects of the PBASS intervention on the composition of gut microbiota, we performed LEfSe analyses using the relative abundance of fecal bacteria to identify the differential taxa as markers representing the baseline and the 2-month and 4-month time points following the initiation of the intervention. The histogram (Figure 2c) shows that there were 28 differential bacterial taxa at the baseline and at the 2- and 4-month time points of the intervention (13, 13, and 2, respectively), all of which had log10 LDA scores above 3.0. The cladogram was then generated to directly visualize and compare the phylogenetic distribution within the baseline and after 2 and 4 months of intervention (Figure 2d).

We further analyzed the proportions of these 28 differential bacterial taxa obtained from LEfSe analyses. After adjusting for age, gender, and PBASS compliance, the older adults were found to have significantly decreased levels of the following bacterial taxa: class Bacilli, genus *Streptococcus*, genus *Ruminiclostridium_5*, class Deltaproteobacteria, phylum Actinobacteria, class Bifidobacteriales, and phylum Patescibacteria and increased levels of genus *Lactobacillus* after 2 and 4 months relative to the baseline (Table 6). There were significant increases in the phylum Bacteroidetes and species *Bacteroides thetaiotaomicron* after 2 months and in the genus *Agathobacter* after 4 months relative to the baseline. There were no appreciable differences in the ratios of Firmicutes and Bacteroidetes—an indicator that is strongly associated with several diseases—at the baseline and 2 and 4 months after the commencement of the intervention (6.6, 4.0, and 6.85, respectively).

To search for the relationships of gut microbiota and TAC in the plasma and erythrocytes, the logarithmically transformed percentages of 28 differential bacterial taxa were included as the dependent variables, using a forward stepwise multiple linear regression analysis. The results showed that the fecal abundances of *Ruminiclostridium_5*, *Agathobacter*, *Bacteroidetes* and *Fusobacteriales* were negatively associated with plasma TAC (*p* = 0.003, R^2^ = 0.929). In addition, the fecal abundance of *Lactobacillus salivarius* was positively associated with erythrocytes TAC (*p* = 0.005, R^2^ = 0.891).

### 3.7. Power and Effect Size Calculation

The calculation of effect size and power using a sample size of 42 or 40 (for fecal samples) in each time point were conducted for the significantly altered primary outcomes, namely indicators of oxidative status and alpha-diversity indices of gut bacterial composition. The effect sizes for plasma GSH, GSSG and TAC were 0.879, 0.425 and 0.618 and erythrocyte GSH, SOD activity and GPx activity were 0.220, 0.561 and 0.316, respectively. In addition, the effect sizes for observed OTUs, Chao 1 and ACE were 0.390, 0.401 and 0.415, respectively. The power to detect these effect sizes was over 0.950 and the sample size needed for these primary outcomes was within the range of 12 to 53 in each time point.

## 4. Discussion

Dietary intake influences the aging process. Diets that are rich in vegetables, fruits, whole grains, and nuts have been found to have anti-oxidative, anti-inflammatory, and anti-aging effects [4] and can potentially regulate the gut microbiota and metabolism [5,10,21]. However, older adults commonly consume limited quantities of these plant foods because of changes in their chewing and swallowing functions [8]. In this study, a significant increase in plasma TAC and altered gut microbiota composition occurred in healthy and sub-healthy older adults, who were provided with 4 types of plant-based, antioxidant-rich smoothies and 2 types of sesame seed snacks for a period of 4 months.

It has been reported that women who adhere closely to dietary regimes entailing high proportions of vegetables have relatively long telomeres in their DNA, which may be associated with decreased CRP [6]. There is emerging evidence that increasing dietary diversity may improve blood antioxidant status [5]. To increase plant food consumption and dietary diversity among older adults, we designed 4 types of plant-based smoothies containing different types of antioxidant vitamins and phytochemicals. The dominant phytochemicals in the orange, green, dark green, and purple smoothies were carotenoids, chlorophyll, chlorophyll/glucosinolate, and anthocyanidins/glucosinolate, respectively. In addition, there are high levels of phenolics, flavonoids, and lignans, such as sesamol, sesamin, and sesamolin in black sesame seeds [22]. All of these phytochemicals are associated with strong antioxidant activity.

In this open-label single-arm intervention study, we found that there were notable increases in endogenous antioxidant thiol groups, such as GSH, TSH and PBSH, and TAC, in the plasma of older adults who participated in this study after 2 months, and these levels continued to rise significantly after 4 months (Table 3). These results suggest that PBASS consumption may increase systemic antioxidant ability in a time-dependent manner in older adults. The findings of a systematic review and meta-analysis indicated that consumption of fruit or vegetable juices did not significantly alter antioxidant capacity, SOD, or catalase in the plasma [23]. These non-significant changes may be attributed to the wide range of ages of the participants (20–70 years), their disparate health conditions, and the varied juice composition, dosages, and duration of consumption in the studies that were analyzed. In this study, we recruited older adults who had consistent regular meals, lifestyle, physical activity, dietary supplement and medicine intake and were provided with the same amounts of PBASS for 4 months. Our results demonstrated that PBASS may improve systemic antioxidant ability in older adults after adjustments have been made for age, gender, and compliance of PBASS consumption.

We also observed a significant increase in erythrocytic TSH and PBSH levels after 4 months of PBASS consumption; however, levels of GSH, GSH/GSSG, and SOD and GPx activities decreased significantly during this period (Table 4). The inconsistent changes in the plasma and erythrocytes in terms of antioxidant indices may have been associated with the condition of erythrocytes, as reflected in the decreased levels of RBC and hematocrit (Table 2). From another perspective, the decrease in antioxidant enzyme activities may be associated with the production of fewer free radicals and reactive oxygen species, leading to decreased demand in the enzymatic antioxidant defense system. In a longitudinal study, Rink et al. [23] found that SOD and GPx activity in erythrocytes was inversely associated with servings of fruits and vegetables consumed by premenopausal women. This inverse relationship is consistent with the view that the activation of antioxidant enzymes occurs mainly as a protective response against oxidative stress [24]. Considered together, these results reveal that PBASS may enhance the non-enzymatic antioxidant defense system and the total antioxidant ability in the plasma.

Studies have found that the composition of gut microbiota alters during the human lifespan [7]. For example, high levels of consumption of plant foods within the diet is associated with increased richness and diversity of gut microbiota and beneficial metabolomic profiles, which impact intestinal health, oxidative status, and inflammation [25]. In a large-scale population-based, cross-sectional study, adults who consumed more fruits were found to have higher alpha- and beta-diversity indices [26]. However, significant decreases in the abundance and richness of gut bacterial species, that is, the observed OTUs and indices of Chao1 and ACE were found in the present intervention (Table 5). The alpha-diversity (indices of Shannon and Simpson; Table 5) and beta-diversity (PCoA analysis; Figure 2a) were not significantly affected by PBASS consumption in older adults. The decreased bacterial species richness and unaltered bacterial diversity may partially explain the unaltered fecal levels of SCFAs (Table 5).

However, the relative abundances of particular bacteria were altered. For example, PBASS consumption significantly increased the abundance of phyla Bacteroidetes and *B. thetaiotaomicron*, the dominant bacterial genus within the gut microbial community of long-living elderly individuals [27], and decreased the abundance of Actinobacteria, one of the dominant bacteria taxa in older adults, and Patescibacteria (Table 6), a newly defined superphylum found in groundwater [28], in healthy and sub-healthy older adults. Opportunistic Deltaproteobacteria pathogens within the phylum Proteobacteria, which reportedly increase in older adults [29], were reduced as a result of PBASS consumption. Moreover, PBASS consumption significantly increased the abundance of genera *Agathobactor* and decreased that of class Bacilli and genera *Streptococcus* and *Ruminiclostridium_5* in older adults. Reduced levels of Butyrate-producing *Agathobactor* bacteria have been reported in children with sleep problems and the core symptoms of autism [30], while one study found increased levels of *Ruminiclostridium_5* in mice whose diets were supplemented with resistant starch [31]. Another study reported controversial effects of *Bacillus* and *Streptococcus* on human health [10]. In this study, we found that plasma TAC was negatively associated with *Ruminiclostridium_5*, *Agathobacter*, Bacteroidetes, and Fusobacteriales, and erythrocytic TAC was positively associated with *Lactobacillus_salivarius*. These findings confirm that plant-based, antioxidant-rich, texture-modified snacks made up of highly diverse food sources can contribute to beneficial modulation of the interactions between gut microbiota and antioxidant ability. Taken together, our findings suggest that PBASS consumption may have considerable potential to alter the composition of gut microbiota in a direction that leads to healthier and longer lives.

SCFAs, the main metabolites of dietary fiber and resistant starches produced by gut bacteria, possess several physiological activities which contribute to human health and disease [7]. Myhrstad et al. [21] performed a literature search and found that the fecal levels of SCFAs were significantly increased in subjects consuming at least twice the amount of dietary fiber than the controls. They concluded that dietary fiber has the potential to change the gut microbiota and alter metabolic regulation in humans [21]. In this study, older adults with PBASS for 4 months had no significant alteration in fecal levels of SCFAs. In addition, the SCFA-producing bacteria, such as *Bacteroides* spp., *Bifidobacterium* spp., *Clostridium* spp., *Prevotella* spp., *Roseburia* spp., *Ruminococcus* spp., *Salmonella* spp., *Streptococcus* spp., and *Veillonella* spp. [32], were not significantly altered after 4 months of PBASS consumption. In our study, the regular meals of the 2 senior living facilities provided at least 14 g of dietary fiber per 1000 kcal per day to fulfill the dietary reference intakes (DRIs) of older adults in Taiwan. The fiber content was 2.7 to 4.7 g in 1 serving of plant-based smoothies and 0 to 1.6 g in 1 serving of sesame seed snacks, which provided approximately 10% to 25% of the amounts of daily dietary intake. The small amount of increase in dietary fiber may not have significant impacts on the fecal levels of SCFAs and SCFA-producing bacteria in older adults who consumed enough dietary fiber from the meals.

The present longitudinal study had several strengths and limitations. Its findings offer insights into cause-and-effect relationships between PBASS consumption and antioxidant ability and/or gut microbiota in healthy and sub-healthy older adults. The majority of the ingredients of PBASS were not provided in the meals of the institutes. This could be part of the reasons that contribute to the significances detected in the study. The older adults in our study demonstrated a high compliance rate relating to PBASS consumption, given that the 4 types of smoothies and 2 types of sesame seed snacks were colorful, tasty, and contained different phytochemicals. The convenience and multiple combinations of PBASS, which can be mixed with milk, juice, ice cream, unseasoned porridge and other foods, encourages a commitment toward consistent daily consumption of PBASS among older adults. Moreover, the relationship between blood TAC and gut microbiota was strong (R^2^ = 0.929 and 0.891 for plasma and erythrocytic TACs, respectively). However, the sample used in this study was small and limited, covering only 2 senior living facilities and comprising more women than men. This single-arm intervention has its limitation in the lack of control group as a comparison. Therefore, we used repeated measure analysis of variance with adjustment for age, gender and compliance of PBASS, and compared the results of the intervention with baseline during the study time frame to show the impact of PBASS in older adults. Double-blind, randomized control trials that are well-powered and well-designed are needed to confirm the effects of plant-based, antioxidant-rich snacks on preventing and alleviating the development of AACD.

## 5. Conclusions

The results of the present study demonstrated that highly diverse, plant-based, antioxidant-rich snacks may elevate antioxidant ability and alter the composition of gut microbiota in older adults within a 4-month period. This study provides new insights into the associations of antioxidant ability with specific changes in the gut microbiota in older adults. Our results suggest that texture-modified, plant-based snacks are useful nutrition support to benefit healthy aging via the elevation of antioxidant ability and alteration of gut microbiota.

## Figures and Tables

**Figure 1 nutrients-13-03872-f001:**
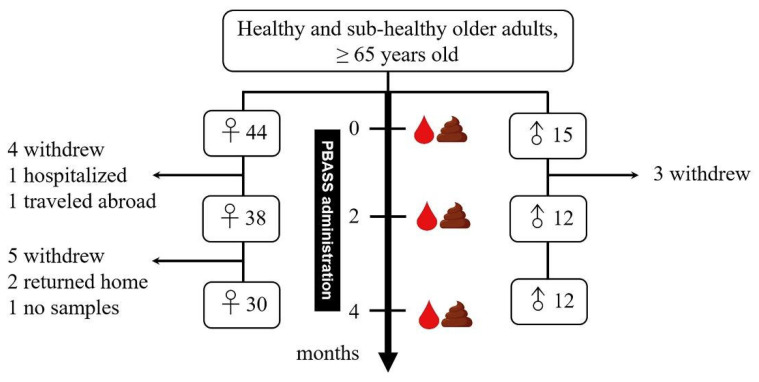
Flow diagram of study participants and study scheme. PBASS: plant-based, antioxidant-rich snacks. Blood and fecal samples were collected at the baseline and after 2 and 4 months of PBASS consumption.

**Figure 2 nutrients-13-03872-f002:**
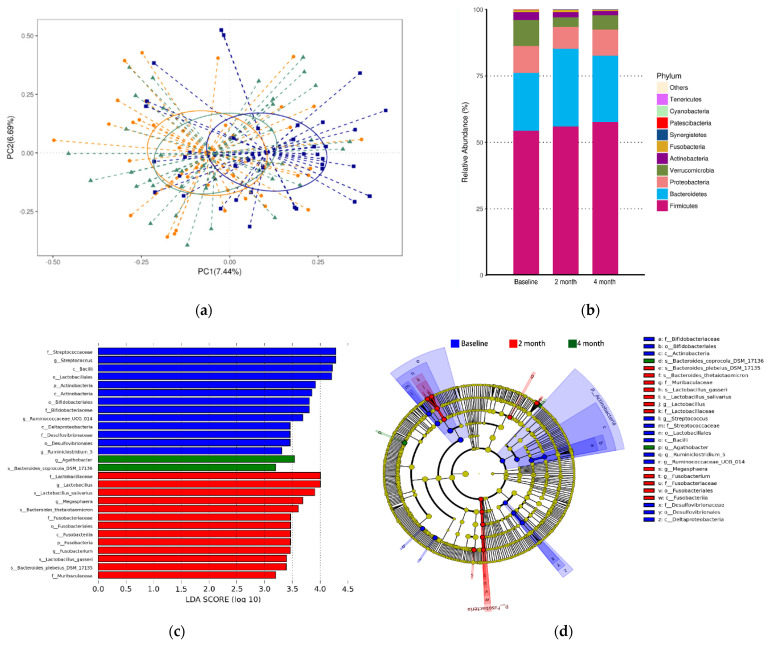
Fecal bacterial composition. (**a**) A score plot obtained from a principal coordinate analysis (PCoA) conducted to evaluate the effects of plant-based, antioxidant-rich smoothies and sesame seed snacks intervention on the composition of gut microbiota. The blue, orange and green spots represent the baseline, 2- and 4-month time points, respectively. Each spot represents a fecal sample. (**b**) The relative abundance of top 10 microorganism populations in feces at phylum level. (**c**) The varying taxa of gut microbiota identified using the linear discriminant analysis (LDA) effect size method. The length of the bar represents the log10 transformed LDA score. Only taxa with absolute LDA (log10) scores >3.0 were considered significant (*p* < 0.05). (**d**) Cladograms predicting the phylogenetic distribution of bacterial lineages associated with the treatment duration. The rings extending from the inner to the outer layers represent different classification levels, namely phylum, class, order, family, genus, and species. The diameter of each circle is proportional to its relative abundance. The names of the taxon levels are abbreviated as follows: p = phylum; c = class; o = order; f = family, g = genus, and s = species.

**Table 1 nutrients-13-03872-t001:** Anthropometric measurements.

Parameters	Baseline	2nd Month	4th Month	Adjusted *p*-Value	Δ2mo	Δ4mo	Δ*p*-Value
Body height (cm)	157.7 ± 7.8	158.4 ± 7.5	157.6 ± 7.2	0.815	0.22 ± 1.63	−0.10 ± 0.78	0.153
Body weight (kg)	59.9 ± 8.8 ^b^	61.1 ± 9.5 ^a^	60.5 ± 9.7 ^ab^	0.028	0.33 ± 2.84	−0.05 ± 1.44	0.219
BMI (kg/m^2^)	24.0 ± 2.8	24.3 ± 3.1	24.3 ± 3.2	0.250	0.00 ± 0.78	−0.16 ± 0.55	0.440
Waist circumference (cm)	85.4 ± 9.0	86.3 ± 10.0	85.8 ± 11.0	0.741	−1.75 ± 2.90	−4.31 ± 4.41	0.830
Hip circumference (cm)	96.7 ± 5.1	97.9 ± 5.5	97.7 ± 11.6	0.132	−0.04 ± 2.15	−6.07 ± 4.44	0.480
Waist-to-hip ratio	0.88 ± 0.07	0.88 ± 0.08	0.88 ± 0.07	0.939	−0.03 ± 0.02	−0.01 ± 0.04	0.264

Values are means ± SD, *n* = 42 (30 females and 12 males). Δ2mo = changes occurring between the baseline and the 2-month time point; Δ4mo = changes occurring between the baseline and the 4-month time point. Values with different superscript letters indicate significant differences within the baseline and 2- and 4-month time points (repeated measure analysis of variance with adjustment for age, gender and compliance of plant-based, antioxidant-rich smoothies and sesame seed snacks and a post hoc analysis using Tukey’s HSD test, *p* < 0.05).

**Table 2 nutrients-13-03872-t002:** Complete blood counts and serum biochemical parameters.

Parameters	Baseline	2nd Month	4th Month	Adjusted *p*-Value	Δ2mo	Δ4mo	Δ*p*-Value
RBC (10^6^/mL)	4.26 ± 0.45 ^a^	4.26 ± 0.50 ^a^	4.17 ± 0.47 ^b^	0.004	0.00 ± 0.18	−0.09 ± 0.16	0.025
Hemoglobin (g/dL)	12.97 ± 1.46 ^ab^	12.88 ± 1.54 ^b^	13.07 ± 1.51 ^a^	0.033	−0.09 ± 0.58	0.10 ± 0.59	0.140
Hematocrit (%)	40.47 ± 4.26 ^a^	41.05 ± 4.80 ^a^	38.78 ± 4.27 ^b^	<0.001	0.58 ± 1.88	−1.69 ± 1.76	<0.001
Albumin (g/dL)	5.03 ± 1.14 ^a^	4.44 ± 0.38 ^b^	4.17 ± 0.33 ^c^	<0.001	−0.59 ± 1.12	−0.86 ± 1.15	0.285
Cholesterol (mg/dL)	185.1 ± 42.0	191.9 ± 46.4	191.8 ± 37.5	0.334	6.8 ± 33.4	6.7 ± 30.3	0.995
LDL-C (mg/dL)	102.0 ± 34.7	110.3 ± 38.3	103.2 ± 30.0	0.076	8.4 ± 23.6	1.3 ± 20.3	0.143
HDL-C (mg/dL)	61.2 ± 12.1 ^b^	68.4 ± 12.4 ^a^	61.8 ± 14.8 ^b^	<0.001	7.2 ± 7.4	0.6 ± 7.1	<0.001
BUN (mg/dL)	18.12 ± 6.32 ^a^	19.00 ± 8.68 ^a^	16.64 ± 6.01 ^b^	0.032	0.88 ± 4.31	−1.48 ± 4.10	0.012

Values are means ± SD, *n* = 42 (30 females and 12 males). RBC, red blood cell; BUN, blood urea nitrogen; LDL-C, low-density lipoprotein-cholesterol; HDL-C, high-density lipoprotein-cholesterol; Δ2mo = changes occurring between the baseline and the 2-month time point; Δ4mo = changes occurring between the baseline and the 4-month time point. Values with different superscript letters indicate significant differences within the baseline and 2- and 4-month time points (repeated measure analysis of variance with adjustment for age, gender and compliance of plant-based, antioxidant-rich smoothies and sesame seed snacks and a post hoc analysis using Tukey’s HSD test, *p* < 0.05).

**Table 3 nutrients-13-03872-t003:** Plasma concentrations of antioxidant indices and lipid peroxidation products.

Parameters	Baseline	2nd Month	4th Month	Adjusted *p*-Value	Δ2mo	Δ4mo	Δ*p*-Value
GSH (mg/mL)	212.1 ± 28.9 ^c^	334.4 ± 41.2 ^b^	364.0 ± 35.1 ^a^	<0.001	122.3 ± 44.0	151.9 ± 38.2	0.002
GSSG (mg/mL)	36.4 ± 18.4 ^a^	23.0 ± 9.4 ^b^	34.7 ± 8.1 ^a^	<0.001	−13.5 ± 19.9	−1.7 ± 19.9	0.008
GSH/GSSG	6.66 ± 2.01 ^c^	16.48 ± 5.64 ^a^	11.27 ± 4.00 ^b^	<0.001	9.82 ± 5.69	4.61 ± 4.70	<0.001
TSH (nmol/mL)	226.2 ± 46.5 ^ab^	222.4 ± 24.0 ^b^	247.2 ± 45.2 ^a^	0.015	−3.9 ± 46.7	21.0 ± 62.5	0.042
NPSH (nmol/mL)	29.9 ± 3.9	28.1 ± 6.3	29.2 ± 1.7	0.439	−1.8 ± 6.8	−0.7 ± 4.3	0.390
PBSH (nmol/mL)	197.1 ± 45.9 ^ab^	194.3 ± 24.0 ^b^	218.0 ± 45.3 ^a^	0.019	−2.8 ± 46.3	20.9 ± 61.9	0.050
TAC (nmol/mL)	8007 ± 555 ^c^	8443 ± 679 ^b^	9226 ± 691 ^a^	<0.001	436 ± 554	1219 ± 587	<0.001
TBARS (nmol/mL)	7.17 ± 1.40	6.92 ± 1.38	6.93 ± 1.48	0.325	−0.25 ± 1.52	−0.23 ± 1.54	0.962

Values are means ± SD, *n* = 42 (30 females and 12 males). GSH, reduced glutathione; GSSG, oxidized glutathione; GSH/GSSG, ratio of reduced glutathione to oxidized glutathione; TSH, total sulfhydryl groups; NPSH, non-protein sulfhydryl groups; PBSH, protein-bound sulfhydryl groups; TAC, total antioxidant capacity; TBARS, thiobarbituric acid reactive substances; Δ2mo = changes occurring between the baseline and the 2-month time point; Δ4mo = changes occurring between the baseline and the 4-month time point. Values with different superscript letters indicate significant differences within the baseline and 2- and 4-month time points (repeated measure analysis of variance with adjustment for age, gender and compliance of plant-based, antioxidant-rich smoothies and sesame seed snacks and a post hoc analysis using Tukey’s HSD test, *p* < 0.05).

**Table 4 nutrients-13-03872-t004:** Erythrocyte contents of antioxidant indices, lipid peroxidation product and antioxidant enzyme activities.

Parameters	Baseline	2nd Month	4th Month	Adjusted *p*-Value	Δ2mo	Δ4mo	Δ*p*-Value
GSH (mg/10^9^ cell)	464.1 ± 98.0 ^a^	197.8 ± 28.9 ^b^	181.6 ± 22.4 ^c^	<0.001	−266.3 ± 86.5	−282.5 ± 98.2	0.425
GSSG (mg/10^9^ cell)	140.3 ± 85.8 ^a^	65.2 ± 11.9 ^b^	69.3 ± 23.9 ^b^	<0.001	−75.1 ± 85.7	−71.1 ± 85.8	0.830
GSH/GSSG	4.21 ± 0.26 ^a^	3.09 ± 0.07 ^b^	2.85 ± 0.13 ^b^	<0.001	−1.13 ± 0.29	−1.37 ± 0.29	0.559
TSH (nmol/10^9^ cell)	2.18 ± 0.57 ^b^	2.35 ± 0.54 ^b^	2.69 ± 0.55 ^a^	0.004	0.17 ± 0.77	0.50 ± 0.83	0.061
NPSH (nmol/10^9^ cell)	0.14 ± 0.036	0.12 ± 0.03	0.13 ± 0.03	0.064	−0.01 ± 0.03	−0.01 ± 0.03	0.591
PBSH (nmol/10^9^ cell)	2.04 ± 0.57 ^b^	2.23 ± 0.53 ^a^	2.56 ± 0.55 ^a^	0.003	0.18 ± 0.78	0.51 ± 0.83	0.065
TAC (nmol/10^9^ cell)	2327 ± 355	2357 ± 361	2292 ± 457	0.526	30 ± 324	−35 ± 502	0.487
TBARS (nmol/10^9^ cell)	48.54 ± 13.73	47.48 ± 11.51	49.27 ± 10.43	0.494	−1.06 ± 11.16	0.73 ± 10.76	0.459
SOD	45.56 ± 8.62 ^a^	36.20 ± 6.45 ^b^	35.24 ± 5.41 ^b^	<0.001	−9.35 ± 6.74	−10.32 ± 7.31	0.530
GPx	172.4 ± 31.5 ^a^	168.3 ± 34.7 ^a^	149.2 ± 24.9 ^b^	<0.001	−4.0 ± 24.8	−23.1 ± 20.7	<0.001
Catalase	0.72 ± 0.62	0.55 ± 0.33	0.89 ± 1.02	0.052	−0.16 ± 0.68	0.17 ± 1.22	0.128

Values are means ± SD, *n* = 40 (29 females and 11 males). OTUs, operational taxonomic units; Chao1, Chao richness estimator 1; ACE, abundance-based coverage estimator; Δ2mo = changes occurring between the baseline and the 2-month time point; Δ4mo = changes occurring between the baseline and the 4-month time point. Values with different superscript letters indicate significant differences within the baseline and 2- and 4-month time points (repeated measure analysis of variance with adjustment for age, gender and compliance of plant-based, antioxidant-rich smoothies and sesame seed snacks and a post hoc analysis using Tukey’s HSD test, *p* < 0.05).

**Table 5 nutrients-13-03872-t005:** Short-chain fatty acids and microbiome species richness and diversity in the feces.

Parameters	Baseline	2nd Month	4th Month	Adjusted *p*-Value	Δ2mo	Δ4mo	Δ*p*-Value
Acetic acid (μmol/g)	40.98 ± 16.83	38.59 ± 17.86	30.10 ± 17.26	0.179	−2.39 ± 13.00	−4.89 ± 12.92.	0.968
Propionic acid (μmol/g)	40.88 ± 19.21	42.26 ± 20.48	38.89 ± 20.85	0.218	1.37 ± 17.72	−2.00 ± 21.14	0.275
Butyric acid (μmol/g)	36.06 ± 18.20	31.61 ± 16.76	34.21 ± 19.06	0.525	−4.46 ± 21.9	−1.85 ± 20.71	0.711
Observed OTUs	319.5 ± 98.4 ^a^	243.4 ± 71.1 ^b^	252.4 ± 69.4 ^b^	<0.001	−76.0 ± 14.4	−67.1 ± 17.4	0.254
Chao1	391.1 ± 112.5 ^a^	301.2 ± 85.4 ^b^	310.2 ± 77.9 ^b^	<0.001	−89.9 ± 17.0	−80.9 ± 19.4	0.412
ACE	387.9 ± 111.4 ^a^	294.6 ± 78.9 ^b^	308.8 ± 77.9^b^	<0.001	−93.3 ± 16.3	−79.1 ± 19.8	0.225
Shannon	5.09 ± 0.74	4.98 ± 0.68	5.00 ± 0.66	0.298	−0.11 ± 0.13	−0.08 ± 0.12	0.596
Simpson	0.92 ± 0.06	0.92 ± 0.05	0.93 ± 0.04	0.425	0.00 ± 0.01	0.01 ± 0.01	0.596

Values are means ± SD, *n* = 42 (30 females and 12 males). GSH, reduced glutathione; GSSG, oxidized glutathione; GSH/GSSG, ratio of reduced glutathione to oxidized glutathione; TSH, total sulfhydryl groups; NPSH, non-protein sulfhydryl groups; PBSH, protein-bound sulfhydryl groups; TAC, total antioxidant capacity; TBARS, thiobarbituric acid reactive substances; SOD, superoxide dismutase; GPx, glutathione peroxidase; Δ2mo = changes occurring between the baseline and the 2-month time point; Δ4mo = changes occurring between the baseline and the 4-month time point. Units: SOD, unit/mg of hemoglobin; GPx, nmol NADPH/min/mg of hemoglobin; catalase, nmol H_2_O_2_/min/mg of hemoglobin. Values with different superscript letters indicate significant differences within the baseline and 2- and 4-month time points (repeated measure analysis of variance with adjustment for age, gender and compliance of plant-based, antioxidant-rich smoothies and sesame seed snacks and a post hoc analysis using Tukey’s HSD test, *p* < 0.05).

**Table 6 nutrients-13-03872-t006:** The abundances of gut microbiota.

Parameters	Baseline	2nd Month	4th Month	Adjusted *p*-Value	Δ2mo	Δ4mo	Δ*p*-Value
p_Firmicutes (%)	54.30 ± 20.15	55.95 ± 15.63	57.59 ± 13.77	0.333	1.65 ± 19.88	3.29 ± 22.36	0.730
c_Bacilli	7.18 ± 9.41 ^a^	4.20 ± 10.92 ^b^	4.78 ± 8.38 ^b^	<0.001	−2.98 ± 11.75	−2.40 ± 10.79	0.818
o_Lactobacillales	6.68 ± 11.32	3.78 ± 8.45	5.55 ± 8.84	0.451	−2.90 ± 14.35	−1.13 ± 14.93	0.591
f_Streptococcaceae	4.12 ± 7.61	4.14 ± 6.42	1.63 ± 2.49	0.234	0.02 ± 8.51	−2.49 ± 8.02	0.179
g_Streptococcus	0.057 ± 0.087 ^a^	0.018 ± 0.03 ^b^	0.024 ± 0.039 ^b^	0.003	−0.040 ± 0.089	−0.033 ± 0.095	0.765
f_Lactobacillaceae	1.088 ± 3.215	2.364 ± 8.976	1.733 ± 6.439	0.745	1.276 ± 9.511	0.644 ± 7.430	0.742
g_Lactobacillus	0.009 ± 0.026 ^b^	0.023 ± 0.097 ^a^	0.020 ± 0.056 ^a^	0.005	0.013 ± 0.098	0.010 ± 0.056	0.869
s_Lactobacillus_salivarius	0.003 ± 0.012	0.013 ± 0.079	0.012 ± 0.040	0.543	0.010 ± 0.078	0.009 ± 0.040	0.931
s_Lactobacillus_gasseri	0.002 ± 0.008	0.006 ± 0.030	0.003 ± 0.011	0.231	0.004 ± 0.031	0.001 ± 0.012	0.539
g_Ruminiclostridium_5	0.008 ± 0.011 ^a^	0.004 ± 0.006 ^b^	0.004 ± 0.012 ^b^	0.003	−0.004 ± 0.013	−0.003 ± 0.017	0.905
g_Ruminococcaceae_UCG_014	0.016 ± 0.029	0.009 ± 0.019	0.009 ± 0.019	0.165	−0.007 ± 0.022	−0.007 ± 0.032	0.972
g_Agathobacter	0.014 ± 0.030 ^b^	0.017 ± 0.024 ^b^	0.019 ± 0.028 ^a^	0.022	0.003 ± 0.036	0.005 ± 0.039	0.854
g_Megasphaera	0.012 ± 0.026	0.020 ± 0.049	0.010 ± 0.029	0.963	0.008 ± 0.056	−0.002 ± 0.041	0.364
p_Bacteroidetes (%)	21.80 ± 17.26 ^b^	29.23 ± 15.19 ^a^	25.01 ± 13.12 ^ab^	0.031	7.43 ± 17.45	3.21 ± 18.39	0.296
f_Muribaculaceae	0.470 ± 1.099	0.559 ± 1.305	0.375 ± 0.857	0.698	0.089 ± 1.190	−0.095 ± 1.467	0.541
s_Bacteroides_plebeius_DSM_17135	0.008 ± 0.016	0.009 ± 0.029	0.005 ± 0.014	0.575	0.002 ± 0.029	−0.002 ± 0.014	0.449
s_Bacteroides_thetaiotaomicron	0.006 ± 0.008 ^b^	0.015 ± 0.025 ^a^	0.009 ± 0.012 ^ab^	0.017	0.009 ± 0.024	0.004 ± 0.014	0.228
s_Bacteroides_coprocola_DSM_17136	0.002 ± 0.010	0.003 ± 0.009	0.005 ± 0.016	0.879	0.000 ± 0.014	0.003 ± 0.019	0.560
p_Proteobacteria (%)	10.16 ± 9.33	8.32 ± 9.21	9.77 ± 10.38	0.160	−1.84 ± 11.54	−0.38 ± 11.32	0.571
c_Deltaproteobacteria	0.909 ± 1.460 ^a^	0.372 ± 0.414 ^b^	0.413 ± 0.549 ^b^	0.006	−0.537 ± 1.520	−0.497 ± 1.629	0.908
o_Desulfovibrionales	0.488 ± 0.647	0.473 ± 0.607	0.733 ± 1.399	0.756	−0.015 ± 0.843	0.245 ± 1.188	0.263
f_Desulfovibrionaceae	0.733 ± 1.399	0.488 ± 0.647	0.473 ± 0.607	0.745	−0.245 ± 1.454	−0.260 ± 1.114	0.959
p_Actinobacteria (%)	3.04 ± 4.30 ^a^	1.92 ± 4.22 ^b^	1.69 ± 3.36 ^b^	0.001	−1.13 ± 5.65	−1.36 ± 4.76	0.845
c_Actinobacteria	1.947 ± 3.674 ^a^	1.255 ± 4.070 ^b^	0.759 ± 1.671 ^b^	<0.001	−0.692 ± 5.266	−1.188 ± 3.776	0.629
o_Bifidobacteriales	1.06 ± 2.83	0.85 ± 1.77	1.71 ± 4.63	0.656	−0.22 ± 3.25	0.65 ± 5.44	0.393
f_Bifidobacteriaceae	1.708 ± 4.630	1.063 ± 2.826	0.847 ± 1.774	0.749	−0.645 ± 5.683	−0.862 ± 4.883	0.856
p_Fusobacteria (%)	0.648 ± 2.908	0.726 ± 3.284	0.934 ± 3.071	0.210	0.077 ± 4.200	0.286 ± 3.060	0.619
o_Fusobacteriales	0.182 ± 0.519	0.574 ± 3.255	0.316 ± 1.118	0.364	0.392 ± 3.319	0.134 ± 3.154	0.620
f_Fusobacteriaceae	0.093 ± 0.371	0.018 ± 0.052	0.057 ± 0.326	0.104	−0.075 ± 0.309	−0.036 ± 0.459	0.656
g_Fusobacterium	0.006 ± 0.028	0.007 ± 0.033	0.003 ± 0.011	0.198	0.001 ± 0.041	−0.003 ± 0.030	0.660
p_Patescibacteria (%)	0.135 ± 0.299 ^a^	0.025 ± 0.054 ^b^	0.016 ± 0.020 ^b^	0.011	−0.110 ± 0.295	−0.119 ± 0.303	0.899

Values are means ± SD, *n* = 40 (29 females and 11 males). p = phylum; c = class; o = order; f = family; g = genus; s = species; Δ2mo = changes occurring between the baseline and the 2-month time point; Δ4mo = changes occurring between the baseline and the 4-month time point. Values with different superscript letters indicate significant differences within the baseline and 2- and 4-month time points (repeated measure analysis of variance with adjustment for age, gender and compliance of plant-based, antioxidant-rich smoothies and sesame seed snacks and a post hoc analysis using Tukey’s HSD test, *p* < 0.05).

## Data Availability

The data presented in this study are available on request from the corresponding author.

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
