# Peer review of "Plant-Based, Antioxidant-Rich Snacks Elevate Plasma Antioxidant Ability and Alter Gut Bacterial Composition in Older Adults"

_nutrients, 2021, doi:10.3390/nu13113872_

Round 1

Reviewer 1 Report

This is a very interesting study that I enjoyed reading which clearly demonstrates that improved nutrition can support healthy ageing via antioxidant properties. The paper was very well written and easy to read. It also validated the regular addition of smoothies to my own diet as a slightly younger individual!

My only comments relate to two things

  • section 2.3 methods. 

Do the authors have more details they can provide about the composition of their orange, green, dark green and purple smoothies. All smoothies contained two kinds of fruit and two kinds of vegetables, but I would have liked more info about these ie contained banana, rockmelon, or? spinach (green), avocado, celery, carrot(orange)  or beetroot (red). I feel that this information is useful in interpreting the results. I also would have liked the % fat ( or more information on type) or grams of fat, and then info on the exchange of nuts was it walnut, peanut or hazelnut. Or is 7 g lipid mean 7g  fat. It is also unclear if they were dairy free or contained a plant-based milk.  Thus in the smoothies where was the lipid coming from? Was it a controlled source – ie a particular oil- or eg if coconut milk added this would be a different fat to milk

In addition I looked at the reference given for this section 14- which just had dietary assessments on patients and I cannot see the relevance of it to this method section. I expected to see a paper that gave more details on the composition of the snacks provided. Can the authors please check this is the appropriate reference.

Shikany, J.M.; Demmer, R.T.; Johnson, A.J.; Fino, N.F.; Meyer, K.; Ensrud, K.E.; Lane, N.E.; Orwoll, E.S.; Kado, D.M.; Zmuda, 558 J.M., et al. Association of dietary patterns with the gut microbiota in older, community-dwelling men. Am J Clin Nutr 2019, 110, 559 1003-1014, doi:10.1093/ajcn/nqz174

I guess the results reported do not compare functional responses to the drinks selected but in some ways given that participants had a choice of smoothies and would have had their own personal preferences I would have thought that the different coloured smoothies would actually provide different minerals, vitamins, and phytochemicals and also types of fiber. Beetroot in particular is an excellent source of inulin which would have strong effects on the microbiota.

Given this diversity in smoothies and perhaps sesame snacks consumed in someways it is surprising that the effects of the dietary intervention were able to be detected with significance. Can the authors comment on this?

A major flaw of this work to me is that how do we know that it is the smoothies that made the difference to participants and not some major change in the regular food provided by the institution, as this may had a stronger effect than anything else. Why was there not a control group for this study. Ie residents that did not go on the smoothies but consumed the food at the institution for the 4 months of the study.

  • SCFA analysis.  The fibre content of smoothies was 2.7 to 4.7 g and it is not stated for the sesame seed powder. I am assuming this would mostly be insoluble fiber coming from the fruit and vegetables.  SCFA were detected but not altered during the 4-month intervention. Can the authors comment on whether  in other studies in humans this amount of insoluble fibre being added to the diet has resulted in increases in SCFA production.  Also did the authors specifically look to see if there were changes in the specific bacteria that can produce SCFA, to see if their abundance data matched their SCFA result.

Reviewer 2 Report

In my opinion it is a very interesting and well conducted study with an interesting results which give some opportunities for further investigations. The topic of the manuscript fits well to the scope of the Journal. A several comments and suggestions included below:

Line 44:  „against the progression of aging” – aging is happening anyway, rewrite the sentence, maybe against aging related disorders?

Line 118: how much g the serving has? You can provide it in the bracket as an additional information.

Line 269-270: Looking to the results in a Table 2 I am not sure whether we can conclude about significant increase of haemoglobin and a differences are very little a specially taking into account a standard deviation.
